# OVOR: OnePrompt with Virtual Outlier Regularization for Rehearsal-Free Class-Incremental Learning

**Wei-Cheng Huang, Chun-Fu (Richard) Chen, Hsiang Hsu**
Global Technology Applied Research, JPMorgan Chase, USA
{wei-cheng.huang,richard.cf.chen,hsiang.hsu}@jpmchase.com

## ABSTRACT

Recent works have shown that by using large pre-trained models along with learnable prompts, rehearsal-free methods for class-incremental learning (CIL) settings can achieve superior performance to prominent rehearsal-based ones. Rehearsal-free CIL methods struggle with distinguishing classes from different tasks, as those are not trained together. In this work we propose a regularization method based on virtual outliers to tighten decision boundaries of the classifier, such that confusion of classes among different tasks is mitigated. Recent prompt-based methods often require a pool of task-specific prompts, in order to prevent overwriting knowledge of previous tasks with that of the new task, leading to extra computation in querying and composing an appropriate prompt from the pool. This additional cost can be eliminated, without sacrificing accuracy, as we reveal in the paper. We illustrate that a simplified prompt-based method can achieve results comparable to previous state-of-the-art (SOTA) methods equipped with a prompt pool, using much less learnable parameters and lower inference cost. Our regularization method has demonstrated its compatibility with different prompt-based methods, boosting those previous SOTA rehearsal-free CIL methods' accuracy on the ImageNet-R and CIFAR-100 benchmarks. Our source code is available at https://github.com/jpmorganchase/ovor.

## 1 INTRODUCTION

Continual learning, with its aim to train a model across a sequence of tasks while mitigating catastrophic forgetting as initially identified by McCloskey & Cohen (1989), addresses the challenge of severe performance degradation on previously-seen data when exposed to new samples. Within the continual learning paradigm, class-incremental learning (CIL) involves presenting the model with sequential tasks, each containing samples from different classes, with the ultimate goal of building a universal model spanning all seen classes across tasks. Continual learning algorithms can be broadly categorized into two main streams: rehearsal-based and rehearsal-free methods. Rehearsal-based algorithms store subsets of data from previous tasks for model fine-tuning, but this approach may not always be practical due to storage limitations and privacy concerns. In contrast, rehearsal-free algorithms manipulate model weights to encourage the recall of previously-seen samples through regularization techniques. Recent advancements in prompt-based rehearsal-free methods for CIL, including works by Wang et al. (2022c), Smith et al. (2023a), and Wang et al. (2022b), have showcased that by leveraging parameter-efficient fine-tuning (PEFT) techniques introduced by Lester et al. (2021), Hu et al. (2022), and Li & Liang (2021), these methods can achieve comparable, and sometimes even superior, results when compared to their rehearsal-based counterparts.

Despite the notable achievements of prompt-based rehearsal-free methods, they face two significant challenges. First, these methods require the model to make output predictions among all available class labels without the explicit task information, leading to inter-task confusion when evaluating on previously seen samples from multiple tasks. This confusion arises because neural networks tend to assign high output scores to outlier inputs (Nguyen et al., 2015), resulting in the model misclassifying classes from different tasks. From the model's perspective, all data from different tasks can be seen as outliers since they have never been presented together before. Second, prompt-based

approaches often employ a pool of learnable prompts that are inserted into the pre-trained model. Additionally, subsets of this prompt pool are allocated to different tasks, following the parameter isolation concept (Lange et al., 2022), to prevent knowledge interference between tasks. However, this setup necessitates a mechanism for selecting or combining specific subsets of the prompt pool to generate the appropriate prompt. The methods discussed earlier typically use the input's encoding by the pre-trained model as the query, incurring additional computational costs. Moreover, there is uncertainty regarding whether the correct task can be reliably identified from the query, particularly in the context of class-incremental learning (CIL), where task-to-class assignments are random and lack semantic meaning.

In this paper, we aim to address these two challenges at once. We propose a regularization method with virtually-synthesized outliers to reduce the inter-task fusion under the rehearsal-free setting, leading to better performance; while revisiting the necessities of using a complicated mechanism to compose prompt from a prompt pool, and propose a simple prompt-based CIL that leads to better efficiency in both computations and parameters.

Our contributions can be summarized as follow:

1. We propose an outlier regularization technique that tightens the decision boundaries of the classes among different tasks, and serves as an additional regularization term that can be applied to various prompt-based CIL methods to promote the performance thereof.

2. We demonstrate that for rehearsal-free CIL, prompt-based method using one single prompt, named OnePrompt, can obtain comparable results with previous state-of-the-art (SOTA) prompt-based methods using much more prompts, leading to a far faster inference speed and less number of trainable parameters.

3. We propose OVOR, combining OnePrompt and the Virtual Outlier Regularization technique, with extensive evaluations on ImageNet-R and CIFAR-100. Empirical evidence shows that our method can reach performance on par with or superior to existing SOTA prompt-based methods in terms of computations, parameters and accuracy.

## 2 RELATED WORK

**Continual Learning.** For addressing catastrophic forgetting in continual learning, existing approaches are often categorized (Lange et al., 2022; Mai et al., 2022) into regularization-based, replay-based and parameter isolation. Regularization-based methods usually apply regularization during training that encourages similarity between the new and old models, whether in model weights (Kirkpatrick et al., 2016; Zenke et al., 2017; Aljundi et al., 2018) or in model outputs (Li & Hoiem, 2018; Hou et al., 2018; Cha et al., 2020). As our proposed regularization method does not require an old model that the current model needs to compare with, we avoid the additional cost in storage and computation that it entails.

Replay-based methods, also known as rehearsal-based methods, have a replay buffer that stores data from previous tasks, which can be used with current task data to train the model (Rebuffi et al., 2017; Chaudhry et al., 2019; Buzzega et al., 2020; Prabhu et al., 2020; Cha et al., 2021; Chen et al., 2023). Compared to rehearsal-free methods, the shortcomings of replay include the need for a large buffer to reach reasonable accuracy, and the privacy and availability issues in storing past data. One variant of replay, sometimes called pseudo-replay, learns a generative model from past data, and samples data from it in lieu of the replay buffer (Ostapenko et al., 2019; van de Ven et al., 2020). Although they ameliorate the storage problem, the privacy concern persists because of the possibility that the generative model can reveal past data.

Parameter isolation methods dedicate different subsets of model parameters to different tasks (Rusu et al., 2016; Serrà et al., 2018; Ebrahimi et al., 2020; Wang et al., 2020). Either the size of the model grows as more tasks come in, or less parameters will be reserved for each task. This line of approach can be natural solutions for situations when the correct task is known beforehand, otherwise a way to distinguish the tasks will be needed.

**Rehearsal-Free Class-Incremental Learning.** Inspired by works in parameter-efficient fine-tuning (PEFT) with large pre-trained models (Lester et al., 2021; Hu et al., 2022; Li & Liang, 2021),

prompt-based methods (Wang et al., 2022c;b; Smith et al., 2023a) combine PEFT with parameter isolation for rehearsal-free CIL. A fixed pre-trained ViT encoder is used along with a pool of small learnable prompts, where subsets of the pool are reserved for individual tasks, in the fashion of parameter isolation. These prompt-based methods have shown to achieve SOTA performance, as the good generalizability provided by the pre-trained model mitigate the need for rehearsal. One recent work called S-Prompt (Wang et al., 2022a) also generally follows the aforementioned prompt-based methods, but is designed for the domain-incremental scenario, instead of CIL.

On top of prompt-based methods, there are also recent works (Zhou et al., 2023; Ma et al., 2023) exploring the usage of a prototype classifier. Their classifier heads consists of prototypes for each class, and the class whose prototype is closest to the encoding vector of a given input will be the prediction. They leverage fixed pre-trained models similar to prompt-based methods, and can also achieve comparable performance. Two proposed methods under this direction that are contemporaneous to our work are SLCA (Zhang et al., 2023), which stores one Gaussian distribution for each task instead of class encoding mean, and RanPAC (McDonnell et al., 2023), which employs random projections and a Gram matrix for the prototype classifier.

**Class-Incremental Learning with Task Prediction.** As discussed before, one main bottleneck for parameter isolation methods to solve CIL is to identify the correct task for the input. There are works that classify tasks explicitly (von Oswald et al., 2020; Rajasegaran et al., 2020; Abati et al., 2020; Henning et al., 2021), some training a separate network, while others use entropy for the prediction. Kim et al. (2022) further establishes the connection between task prediction and outlier detection, showing that training outlier detectors can help class-incremental learning. However, as it has separate detectors for each task, it requires a replay buffer to calibrate them. For these methods that try to predict the task, they either require rehearsal, or have inferior performance.

## 3 PREREQUISITES

### 3.1 CLASS-INCREMENTAL LEARNING

In the CIL setting of classification, a dataset $\mathcal{D} = (\boldsymbol{x}_i, y_i)_{i=1}^n$, where $\boldsymbol{x}_i \in \mathbb{R}^m$ and $y_i \in \mathcal{Y}$, is split into $T$ tasks, each with disjoint sets of classes, and usually each task contains the same number of classes ($C_t$). Thus, the training and test set for each task are denoted as $\mathcal{D}_{\text{train}}^t, \mathcal{D}_{\text{test}}^t, t \in \{1, \cdots, T\}$. For a classification model, it consists of a feature encoder, $f_{\boldsymbol{\theta}}$ and a classifier head, $h_{\boldsymbol{\phi}}$ which are parameterized by $\boldsymbol{\theta}$ and $\boldsymbol{\phi}$, respectively. Then for a given input $\boldsymbol{x}$, the feature encoder projects it into the feature space $\mathbb{R}^d$ to produce the feature vector $\boldsymbol{z}$, then the head produces the prediction scores $\hat{\boldsymbol{y}}$ corresponding to all classes given $\boldsymbol{z}$, and the class with the highest score becomes the prediction:

$$\boldsymbol{z} = f_{\boldsymbol{\theta}}(\boldsymbol{x}) \in \mathbb{R}^d, \quad \hat{\boldsymbol{y}} = h_{\boldsymbol{\phi}}(\boldsymbol{z}) \in \mathbb{R}^{(C_t \times T)}, \tag{1}$$

We use superscripts $t$ to indicate the model trained after task $t$, such that the randomly initialized model before training is represented by $(f_{\boldsymbol{\theta}^0}, f_{\boldsymbol{\phi}^0})$, and $(f_{\boldsymbol{\theta}^t}, f_{\boldsymbol{\phi}^t})$ are the model after updating $(f_{\boldsymbol{\theta}^{t-1}}, f_{\boldsymbol{\phi}^{t-1}})$ on the data of task $t$. Let $\mathcal{A}$ denote the learning algorithm (*e.g.*, stochastic gradient descent) that updates the model, such that for each task, given the model trained after task $t-1$, the training data $\mathcal{D}_{\text{train}}^t$ and a training loss $\mathcal{L} : \mathbb{R}^{(C_t \times T)} \times \mathcal{Y} \to \mathbb{R}^+$, the CIL can be formulated as:

$$(\boldsymbol{\theta}^t, \boldsymbol{\phi}^t) = \mathcal{A}((f_{\boldsymbol{\theta}^{t-1}}, f_{\boldsymbol{\phi}^{t-1}}), \hat{\mathcal{D}}_{\text{train}}^t, \mathcal{L}), \tag{2}$$

where $\hat{\mathcal{D}}_{\text{train}}^t = \mathcal{D}_{\text{train}}^t$ for rehearsal-free methods, *i.e.*, only new data alone is available during the training; on the other hand, for rehearsal-based methods, $\hat{\mathcal{D}}_{\text{train}}^t$ can be any subset of the union of all encountered data till now: $\hat{\mathcal{D}}_{\text{train}}^t \subset \bigcup_{t'=1}^t \mathcal{D}_{\text{train}}^{t'}$. We use $\boldsymbol{\theta}/\boldsymbol{\phi}$ and $f_{\boldsymbol{\theta}}/h_{\boldsymbol{\phi}}$ interchangably to denote the feature encoder and classifier head, respectively.

### 3.2 PROMPT-BASED CLASS-INCREMENTAL LEARNING

In this section we outline recent prompt-based CIL methods, in particular L2P (Wang et al., 2022c), DualPrompt (Wang et al., 2022b), and CODA-P (Smith et al., 2023a).

Given the fixed pre-trained ViT encoder $f_{\boldsymbol{\theta}^0}$, classifier head $h_{\boldsymbol{\phi}}$, and the prompt pool containing pairs of keys and prompts $\mathcal{P} = \{(\boldsymbol{k}_m, \boldsymbol{p}_m) \mid m \in \{1, \cdots, M\}\}$, where $\boldsymbol{k}_m, \boldsymbol{p}_m \in \mathbb{R}^d$. Let $\mathcal{F}$ be an algorithm that composes the appropriate prompt $\hat{\boldsymbol{p}} \in \mathbb{R}^d$ given query function $q$ and prompt pool $\mathcal{P}$, such that $\hat{\boldsymbol{p}} = \mathcal{F}(q(\boldsymbol{x}), \mathcal{P})$, $e.g.$, in DualPrompt, $q(\cdot) = f_{\boldsymbol{\theta}^0}(\cdot)$ and $\mathcal{F} : \hat{\boldsymbol{p}} = \boldsymbol{p}_i \arg\max_{i \in \{1, \cdots, M\}} q(\boldsymbol{x})^T \boldsymbol{k}_i$ After the $\hat{\boldsymbol{p}}$ is composed, it will be inserted into the encoder $f_{\boldsymbol{\theta}^0}$ along with the input $\boldsymbol{x}$, and we denote it as $f_{\boldsymbol{\theta}^0, \hat{\boldsymbol{p}}}(\boldsymbol{x}) \in \mathbb{R}^d$ (See Section 4.2 in Wang et al. (2022b) for the different types of prompt insertion.).

For prompt-based models, only the prompts $\mathcal{P}$ and the parameters of head $\boldsymbol{\phi}$ are updated, while the weights of encoder part $\boldsymbol{\theta}^0$ stays unchanged throughout the process. Similar to $\boldsymbol{\theta}$ and $\boldsymbol{\phi}$, we use $\mathcal{P}^t$ to denote the updated prompt pool after the $t$-th task. Then, we use $\mathcal{F}_{\mathcal{P}^t}$ as a shorthand of $\mathcal{F}(\cdot, \mathcal{P}^t)$, such that $\hat{\boldsymbol{p}} = \mathcal{F}_{\mathcal{P}^t}(q(\boldsymbol{x}))$. Additionally, in practice the head can be decomposed based on the number of tasks, and each output score for classes that belongs to a particular task. We differentiate them by underscripts, such that $\boldsymbol{\phi} = \{\boldsymbol{\phi}_1, \cdots, \boldsymbol{\phi}_T\}$. Following Wang et al. (2022b), each $\boldsymbol{\phi}_t$ will only be trained on task $t$, in order to avoid recency bias. Thus, the prediction of the classifier head can be expressed as follow:

$$\hat{\boldsymbol{y}} = h_{\boldsymbol{\phi}}(f_{\boldsymbol{\theta}^0, \hat{\boldsymbol{p}}}(\boldsymbol{x})) = [h_{\boldsymbol{\phi}_1}(f_{\boldsymbol{\theta}^0, \hat{\boldsymbol{p}}}(\boldsymbol{x})); \cdots; h_{\boldsymbol{\phi}_T}(f_{\boldsymbol{\theta}^0, \hat{\boldsymbol{p}}}(\boldsymbol{x}))], \quad \text{where } h_{\boldsymbol{\phi}_i}(\cdot) \in \mathbb{R}^{C_t}. \quad (3)$$

Now in the context of prompt-based rehearsal-free CIL, Equation 2 becomes:

$$(\boldsymbol{\phi}_t^t, \mathcal{P}^t) = \mathcal{A}((f_{\boldsymbol{\theta}^0}, h_{\boldsymbol{\phi}_t^{t-1}}, \mathcal{F}_{\mathcal{P}^{t-1}}), \mathcal{D}_{\text{train}}^t, \mathcal{L}). \quad (4)$$

In the above learning process only $\mathcal{P}^t$ and $\boldsymbol{\phi}_t^t$ are updated, where $\boldsymbol{\theta}^0$ and the remaining parts of $\boldsymbol{\phi}$ are not, hence they are not present in the output of $\mathcal{A}$. To conclude, aforementioned prompt-based methods follow the process below:

$$\hat{\boldsymbol{p}}^t = \mathcal{F}(f_{\boldsymbol{\theta}^0}(\boldsymbol{x}), \mathcal{P}^t), \quad \hat{\boldsymbol{y}} = h_{\boldsymbol{\phi}}(f_{\boldsymbol{\theta}^0, \hat{\boldsymbol{p}}^t}(\boldsymbol{x})), \quad (5)$$

## 4 METHOD

For rehearsal-free CIL, one common problem is that the classifer head could easily confuse a data sample from one task with other tasks since no data in different tasks are trained together to differentiate them. We called this phenomenon "inter-task confusion". Fig. 1 (top row) describes an example that even though each classifier trained in each task could achieve high accuracy per task; however, lacking data between tasks results in confidence scores from the classifiers of each task that cannot be directly compared. Even so, during the inference, the model usually picks the class with max score as its prediction. To this end, we proposed a model-agnostic regularization method on the classifier head to reduce the inter-task confusion.

### 4.1 VIRTUAL OUTLIER REGULARIZATION

In rehearsal-based CIL, since there are buffers to save data from other tasks, we could use them to train the model with the current data to avoid for inter-task confusion; however, it is impossible for rehearsal-free CIL. Therefore, we propose generating virtual outliers ($e.g.$, NPOS (Tao et al., 2023)) to further regularize the decision boundary formed by the classifier head. Figure 1 (bottom row) illustrates that, with outliers, the decision boundary for each class becomes more compact such that each classifier head only generates higher scores for the samples whose distribution is close to the training data; therefore, the overall performance of CIL are improved due to the lower inter-task confusion.

Thus, for each task, we generate a set of virtual outliers, $\boldsymbol{v} \sim \mathcal{D}_{\text{Outlier}}^t$, $\boldsymbol{v} \in \mathbb{R}^d$, where $\mathcal{D}_{\text{Outlier}}^t = \mathcal{G}(f_{\boldsymbol{\theta}^0, \hat{\boldsymbol{p}}^t}(\mathcal{D}_{\text{train}}^t))$, where $\mathcal{G}$ is a outlier generator, and use it with the feature vectors of current training data $f_{\boldsymbol{\theta}^0, \hat{\boldsymbol{p}}^t}(\mathcal{D}_{\text{train}}^t)$ to regularize the classifier head. We adopt two hinge loss terms used in Liu et al. (2020) for out-of-distribution detection as regularization to enforce the classifier head ($\boldsymbol{\phi}$) to produce higher confidence of the samples in current task via $\tau_{\text{Current}}$ while to lower the

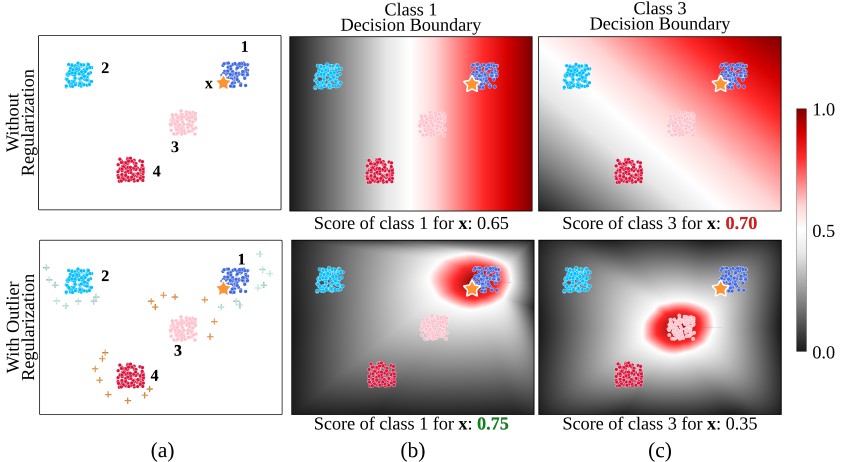

Figure 1: An example of inter-task confusion under rehearsal-free CIL (2-task CIL, each task with 2 classes); class 1 and 2 are in task 1 while class 3 and 4 are in task 2. Each row shows the illustration without and with the proposed regularization. (a) feature vectors for data from all classes, and outliers marked by crosses in the bottom row. (b) and (c) are the decision boundary for class 1 and 3, and the associated scores (We omit class 2 and 4 for simplicity). Given a sample $\boldsymbol{x}$ belonging to class 1, without regularization, during the inference, $\boldsymbol{x}$ is classified as class 3 as the score of class 3 is higher than class 1 since there are no other task data to further confine the decision boundary to avoid inter-task confusion; while, with regularization, the decision boundary become more compact such that the classifier only generates the high scores for those samples that are in the distribution of training data; thus, this time, it gives $\boldsymbol{x}$ higher score for class 1 and correctly classifies it.

confidence of the virtual outliers through $\tau_{\text{Outlier}}$.

$$\mathcal{L}_{\text{Outlier}} = \mathbb{E}_{\boldsymbol{x} \sim \mathcal{D}_{\text{train}}^t} \left[ \mathcal{L}_\delta(\max(0, E_{\text{Current}}(\boldsymbol{x}, t) - \tau_{\text{Current}})) \right]$$
$$+ \mathbb{E}_{\boldsymbol{v} \sim \mathcal{D}_{\text{Outlier}}^t} \left[ \mathcal{L}_\delta(\max(0, \tau_{\text{Outlier}} - E_{\text{Outlier}}(\boldsymbol{v}, t))) \right],$$
$$\text{where } E_{\text{Current}}(\boldsymbol{x}, t) = -\log \sum_j e^{\hat{\boldsymbol{y}}_j^x}, \quad E_{\text{Outlier}}(\boldsymbol{v}, t) = -\log \sum_j e^{\hat{\boldsymbol{y}}_j^v} \quad (6)$$
$$\text{and } \hat{\boldsymbol{y}}^x = h_{\boldsymbol{\phi}_t}(f_{\theta^0, \hat{\boldsymbol{p}}^t}(\boldsymbol{x})) \in \mathbb{R}^{C_t}, \hat{\boldsymbol{y}}^v = h_{\boldsymbol{\phi}_t}(\boldsymbol{v}) \in \mathbb{R}^{C_t},$$

where $\mathcal{L}_\delta(\cdot)$ is the Huber loss (Huber, 1964). Note that we use the Huber loss instead of the mean-square-error (MSE) loss to provide more stable training for better convergence. Note that the outliers are only used to regularize the classifier head $h_\phi$, instead of training separate outlier detectors as in existing outlier synthesis works (Du et al., 2022; Tao et al., 2023). In CIL setting, all tasks are separate, thereby requiring one detector per task; thus, it will be difficult to design an appropriate way to choose the correct task with these detectors, since all of them are trained locally to each individual task. By not having separate detectors, we eliminate the need for introducing and searching for a good set of respective hyperparameters.

In this work, we adopt NPOS (Tao et al., 2023, Section 3) as virtual outlier synthesis and we briefly mention the steps here—for more details, please refer to Appendix D. It first selects the boundary samples as those among input samples with the largest distances with respect to the rest of the input samples. Then, the candidates of virtual outliers are generated by adding randomly sampled noises from a Gaussian distribution $\mathcal{N}(\mathbf{0}, \sigma^2 \boldsymbol{I})$, to the selected boundary samples. After that, for each candidate its distance to the input samples are calculated, and only those with the largest distances are chosen as outliers.

The proposed virtual outlier regularization (VOR) is agnostic to any prompt-based CIL, *e.g.*, CODA-P (Smith et al., 2023a), DualPrompt (Wang et al., 2022b) and L2P (Wang et al., 2022c). In next section, we further explore an efficient prompt-based CIL in terms of FLOPs and parameters, named OnePrompt, which only use a single prompt through the whole CIL process rather than using a complicated mechanism to compose prompt for different tasks.

Table 1: Accuracy difference between the first and final evaluation of test data of each task (*i.e.*, right after each task and task 10) for OnePrompt on 10-task ImageNet-R, given the correct task ID. There is almost no drop, implying little significant representation shift after the prompts are updated.

| Task 1 | Task 2 | Task 3 | Task 4 | Task 5 | Task 6 | Task 7 | Task 8 | Task 9 |
|--------|--------|--------|--------|--------|--------|--------|--------|--------|
| +2.05 | +1.64 | +1.34 | +0.40 | +0.61 | +0.24 | +0.23 | −0.04 | +0.36 |

## 4.2 ONEPROMPT AS PROMPTING METHOD

Despite prompt-based CIL achieved superior performance, the common design is to compose different prompt $\hat{p}$ for different tasks to learn different feature representation because the feature characteristics could vary task by task (Smith et al., 2023a; Wang et al., 2022b;c). Moreover, those methods usually require a query function over the input to compose the prompt since the task index is not given during the index. This bring two drawbacks: first, the query function is usually as complex as the pre-trained model; second, as the classes are randomly shuffled during the training, there is no exact semantic meaning for each task, unlike domain incremental learning whose each task is related to one specific domain. Thus, we first revisit the prompt-based method with a single prompt through the CIL, which means $\mathcal{P} = \{p_1\}$, $\hat{p} = p_1$, and no need for $\mathcal{F}$ since it always outputs the same prompt. We name this method as OnePrompt since only one prompt is used.

Surprisingly, we found that without composing prompts for different tasks, OnePrompt already outperforms L2P and DualPrompt on ImageNet-R with 10-task setting by 4.0% and 1.8% on average accuracy, respectively. Note that, OnePrompt provides $\sim 2\times$ inference speedup because there is no need to compose the prompt from the prompt pool; moreover, it also contains fewer additional parameters to train.

Even though that OnePrompt has delivered good performance, we further investigate whether or not it has the common problem of representation drift when the same set of prompt is continuously updated through all tasks. To validate that, we evaluate OnePrompt on 10-task ImageNet-R, with the correct task index given, so that the factor of inter-task confusion can be eliminated. Table 1 shows performance difference after first and last evaluation, for test data of each task. If there is significant drift in the representation, the corresponding accuracy should be dropping. As one can see, there is almost no degradation, implying little representation drift for OnePrompt. We conducted more analyses to show the representation drift is marginal in Appendix E. We conjecture that since the learnable parameters with OnePrompt is only a small fraction (∼0.3%), the learned feature representation won't drift too much along with the tasks (Smith et al., 2023b).

## 4.3 PROPOSED ONEPROMPT WITH VIRTUAL OUTLIER REGULARIZATION

We proposed OnePrompt with VOR (OVOR) as our prompt-based CIL such that it balances the performance came from OnePrompt and VOR and the efficiency delivered by OnePrompt. Figure 2 illustrates the overview of training procedure of our method. First, for each task $t$, the model is

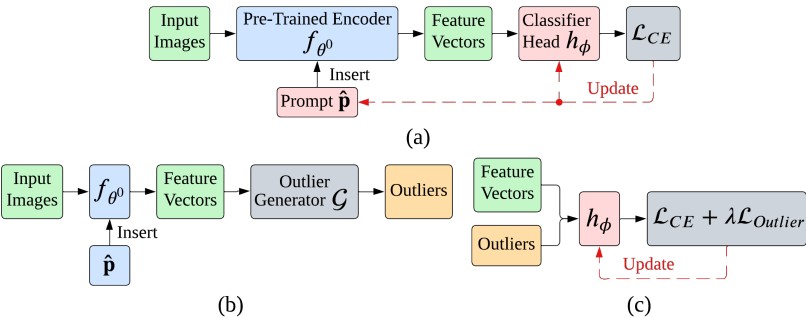

Figure 2: Overview of the training process of proposed OVOR. For each task and the associated training data, three steps are performed to update the model: (a) first, we train the prompt $\hat{p}$ and the head $h_\phi$ with cross-entropy loss, similar to previous methods, where the next steps are the novel parts in OVOR. After model is trained as in (a), the prompt is also frozen as in (b), and all feature vectors of inputs of the current task are computed. These feature vectors are used to generate virtual outlier samples, and in (c) the two sets of vectors are used to update the classifier head $h_\phi$, with both the cross-entropy loss and the proposed regularization loss.

---

**Algorithm 1** OVOR: OnePrompt with Virtual Outlier Regularization

---

**Input:** Pre-trained ViT $f_{\boldsymbol{\theta}^0}$, Initialized prompt $\hat{\boldsymbol{p}}^0$, classifier head $h_{\boldsymbol{\phi}^0}$, Training set $\mathcal{D}_{\text{train}}$
    Learning algorithm $\mathcal{A}$, Loss function $\mathcal{L}_{CE}$, $\mathcal{L}_{\text{Outlier}}$, Outlier generator $\mathcal{G}$, Scalar $\lambda$

    **for** each task $t \in \{1, \cdots, T\}$ **do**
        $(\boldsymbol{\phi}_t^t, \hat{\boldsymbol{p}}^t) \leftarrow \mathcal{A}((f_{\boldsymbol{\theta}^0, \hat{\boldsymbol{p}}^{t-1}}, h_{\boldsymbol{\phi}_t^{t-1}}), \mathcal{D}_{\text{train}}^t, \mathcal{L}_{\text{CE}})$                              $\triangleright$ Fig.2 (a)
        $\mathcal{D}_{\text{Outlier}}^t \leftarrow \{\mathcal{G}(f_{\boldsymbol{\theta}^0, \hat{\boldsymbol{p}}^t}(\boldsymbol{x})) \mid \boldsymbol{x} \in \mathcal{D}_{\text{train}}^t\}$                         $\triangleright$ Fig.2 (b)
        $\boldsymbol{\phi}_t^t \leftarrow \mathcal{A}((f_{\boldsymbol{\theta}^0, \hat{\boldsymbol{p}}^t}, h_{\boldsymbol{\phi}_t^t}), (\mathcal{D}_{\text{train}}^t, \mathcal{D}_{\text{Outlier}}^t), \mathcal{L}_{\text{CE}} + \lambda\mathcal{L}_{\text{Outlier}})$         $\triangleright$ Fig.2 (c)
    **end for**

---

trained with standard cross-entropy loss to learn good feature representation, obtaining updated $\hat{\boldsymbol{p}}^t$ and $h_{\boldsymbol{\phi}^t}$; second, generate virtual outliers $\mathcal{D}_{\text{Outlier}}^t$ from the features $f_{\boldsymbol{\theta}^0, \hat{\boldsymbol{p}}^t}(\boldsymbol{x}), \boldsymbol{x} \in \mathcal{D}_{\text{train}}^t$; third, use the virtual outliers along with the training data to update the head $h_{\boldsymbol{\phi}_t}$ with the combination of cross-entropy loss and the outlier regularization loss. Algorithm 1 also details the training steps of our proposed OVOR.

## 5 EXPERIMENTAL RESULTS

**Datasets.** For evaluation, we use four widely-used datasets, ImageNet-R (Hendrycks et al., 2021a) and CIFAR-100 (Krizhevsky et al., 2009), ImageNet-A (Hendrycks et al., 2021b), CUB-200 (Wah et al., 2011). Among them, CIFAR-100 has 100 classes in total, and other three have 200 classes. Following Wang et al. (2022b), we split the datasets randomly into a number of non-overlapping sets, one for each task, each with the same number of classes. Similar to Smith et al. (2023a), we report numbers for all methods on 5, 10, 20 tasks CIL setting on ImageNet-R, and 10-task CIL setting on CIFAR-100. We test on 10-task setting on CUB-200 and ImageNet-A, following Zhou et al. (2023), also using their train/test split for ImageNet-A. For all experiments, we run 5 times with different random seeds, for the metrics to be described below. We report the mean of 5 runs in Table 2–3, and include the version with standard deviation Table A.5–A.6 in the appendix.

**Metrics.** We use two metrics to measure the performance the methods, including average accuracy ($A_T$), average forgetting ($F_T$). The forgetting metric quantifies the drop in accuracy for a task over time, then averaged over all tasks. For average accuracy and forgetting, we follow the definition in Wang et al. (2022b):

$$A_t = \frac{1}{t} \sum_{i=1}^{t} S_{t,i}, \quad F_t = \frac{1}{t-1} \sum_{i=1}^{t-1} \max_{i' \in \{1, \cdots, t-1\}} (S_{i',i} - S_{t,i}) \tag{7}$$

where $S_{t,i}$ is the evaluation accuracy on $\mathcal{D}_{\text{test}}^t$ for the model after training on task $i$. Note that average accuracy is more important than average forgetting as the $A_T$ already encompasses both plasticity and forgetting, with $F_T$ only providing additional context, as advocated by Smith et al. (2023a). Moreover, we found each paper use different way to compute the average forgetting, so we only report the average forgetting for the methods we reproduced[1].

**Training Details.** For experimental configurations of CIL training, we follow Smith et al. (2023a) as much as possible, using the Adam (Kingma & Ba, 2015) optimizer, and batch size of 128. The model is trained 50 epochs per task on ImageNet-R and 20 for other three datasets. The last 20% of epochs is used for outlier-regularized training when applicable. Similar to Smith et al. (2023a), we adopt the vision transformer-base (ViT-B) checkpoint from Steiner et al. (2022) that pre-trained on ImageNet-21K then fine-tuned on ImageNet-1K (Russakovsky et al., 2015), and a cosine-decaying learning schedule with the initial learning rate of $1e^{-3}$ is used.

For the regularization loss, we set $\lambda$ to 0.1, same as Liu et al. (2020). We use $\tau_{\text{Current}} = -24.0$ and $\tau_{\text{Outlier}} = -3.0$ for the two threshold hyperparameters in the loss term, and $\delta = 1.0$ as in the Huber loss $\mathcal{L}_\delta$, which is a component of the regularization loss. During the generation of virtual outliers, $\alpha \times C_t$ samples are selected as boundary samples from the inputs, and it outputs $\beta \times C_t$ outliers,

---

[1]Despite that Smith et al. (2023a) and Smith et al. (2023b) also reported average forgetting, they were calculating it in a different way than what they reference to.

Table 2: Results on 10-task ImageNet-R, CIFAR-100, CUB-200, ImageNet-A setting. The number in the parenthesis denotes the result with VOR.

| Method | ImageNet-R | | CIFAR-100 | | CUB-200 | | ImageNet-A | |
|---|---|---|---|---|---|---|---|---|
| | $A_T$ | $F_T$ | $A_T$ | $F_T$ | $A_T$ | $F_T$ | $A_T$ | $F_T$ |
| L2P | 69.31 (71.14) | 7.27 (7.10) | 82.43 (83.87) | 6.18 (8.71) | 70.62 (76.33) | 10.90 (9.38) | 40.16 (41.27) | 8.86 (8.79) |
| DualPrompt | 71.47 (73.27) | 6.02 (6.38) | 83.28 (84.85) | 5.75 (7.50) | 71.85 (78.44) | 10.31 (8.23) | 44.85 (45.45) | 10.80 (8.82) |
| CODA-P | 74.87 (76.82) | 6.73 (8.05) | 86.41 (86.53) | 6.39 (9.17) | 72.74 (78.72) | 10.37 (8.73) | 50.20 (51.04) | 8.91 (7.78) |
| OnePrompt | 73.34 (75.61) | 5.36 (5.77) | 85.60 (86.68) | 4.79 (5.25) | 72.19 (78.12) | 10.18 (8.73) | 47.36 (48.49) | 9.75 (7.83) |

Table 3: Results on ImageNet-R (INR), for 5, 10 and 20 tasks, and CIFAR-100, CUB-200, ImageNet-A for 10 tasks.

| Dataset Method | INR-5 | | INR-10 | | INR-20 | | CIFAR-100 | | CUB-200 | | ImageNet-A | |
|---|---|---|---|---|---|---|---|---|---|---|---|---|
| | $A_T$ | $F_T$ | $A_T$ | $F_T$ | $A_T$ | $F_T$ | $A_T$ | $F_T$ | $A_T$ | $F_T$ | $A_T$ | $F_T$ |
| L2 | – | – | 76.06 | – | – | – | – | – | – | – | – | – |
| A-Prompt | – | – | 72.82 | – | – | – | **87.87** | – | – | – | – | – |
| SimpleCIL | 61.28 | – | 61.28 | – | 61.28 | – | 76.21 | – | 85.16 | – | 49.44 | – |
| ADAM | 74.27 | – | 72.87 | – | 70.47 | – | 85.75 | – | **85.67** | – | **53.98** | – |
| L2P | 71.00 | 6.43 | 69.31 | 7.27 | 65.68 | 8.30 | 82.43 | 6.18 | 70.62 | 10.90 | 40.16 | 8.86 |
| DualPrompt | 73.11 | 5.66 | 71.47 | 6.02 | 67.91 | 6.51 | 83.28 | 5.75 | 71.85 | 10.31 | 44.85 | 10.80 |
| CODA-P | 76.29 | 5.92 | 74.87 | 6.73 | 71.70 | 6.62 | 86.41 | 6.39 | 72.74 | 10.37 | 50.20 | 8.91 |
| OnePrompt | 74.65 | 4.60 | 73.34 | 5.36 | 70.42 | 5.57 | 85.60 | 4.79 | 72.19 | 10.18 | 47.36 | 9.75 |
| OnePrompt-Deep | 75.45 | 5.27 | 73.92 | 4.88 | 71.35 | 5.46 | 85.82 | 5.55 | 71.56 | 10.44 | 46.53 | 10.48 |
| OVOR | 76.79 | 4.88 | 75.61 | 5.77 | 73.13 | 6.06 | 86.68 | 5.25 | 78.12 | 8.73 | 48.49 | 7.83 |
| OVOR-Deep | **76.82** | 7.58 | **76.11** | 7.16 | **73.85** | 6.80 | 85.99 | 6.42 | 78.29 | 8.40 | 47.19 | 7.78 |

where $C_t$ is the number of classes per task, and use $K$-NN to approximate the distance. We use $1.0, 10, 160, 100$ for $\sigma$, $\alpha$, $\beta$, and $K$, respectively. These are the configuration for ImageNet-R, for detailed hyperparameters settings for all datasets, please refer to Table C.17 and Table C.18 in the appendix.

For OnePrompt, we adopt prefix tuning to insert prompt, and the prompt is inserted into the first five layers of the ViT model, the prompt length is $(5, 5, 20, 20, 20)$ respectively for the five layers, according to Wang et al. (2022b). Additionally, we test with inserting all layers with same length of 5, same as VPT-Deep in Jia et al. (2022), and we denote it as OnePrompt-Deep. Similarly, OVOR-Deep being OnePrompt-Deep with VOR. We reproduced the results of the compared methods of L2P (Wang et al., 2022c), DualPrompt (Wang et al., 2022b) and CODA-P (Smith et al., 2023a). Please refer to Appendix C for our implementation details.

## 5.1 EFFECTIVENESS OF VIRTUAL OUTLIER REGULARIZATION

We experiment with four different prompt-based methods: L2P (Wang et al., 2022c), DualPrompt (Wang et al., 2022b), CODA-P (Smith et al., 2023a), and OnePrompt, on the aforementioned datasets under the CIL setting. First, in Table 2, we demonstrate the ability of VOR to improve performance for different prompt-based methods. It shows that across all four datasets, all four methods improve their accuracy when they are accompanied by VOR.

One can notice that whether trained with VOR or not, the four methods are ranked the same in performance. Besides that OnePrompt consistently outperforms DualPrompt and L2P, it is also notable that DualPrompt beats L2P. As DualPrompt uses both task-specific and task-agnostic prompts, and L2P has no equivalent shared prompts, this trend further validates our reasoning behind the designing of OnePrompt, as described in Section 4.2.

## 5.2 COMPARISON TO SOTA REHEARSAL-FREE CIL

For more extensive evaluation results, we present the experimental results on all settings of ImageNet-R, CIFAR-100, CUB-200 and ImageNet-A. For comparison, we also include the results from several recent prototype-classifier-based methods: SimpleCIL (Zhou et al., 2023), ADAM (Zhou et al., 2023), and A-Prompt (Ma et al., 2023). Additionally, for L2 (Smith et al., 2023b), which just fine-tunes all attention layers in the pre-trained ViT with L2 regularization. For these works, we simply quote their numbers from the papers.

Table 3 shows results on all datasets. Among results for ImageNet-R, one can observe similar trend as in Table 2, where among the four prompt-based methods, CODA-P always performs the best, fol-

lowed by OnePrompt, DualPrompt, and then L2P. Nonetheless, VOR can improve OnePrompt such that it can outperform CODA-P. OnePrompt-Deep performs better than OnePrompt, and this remains the case when VOR is added, with OVOR-Deep being the best on all three settings. Prototype-classifier-based A-Prompt and ADAM usually beat L2P and DualPrompt, while underperforming CODA-P and OVOR. Generally, as number of tasks increases, accuracy decreases, forgetting increases, and the improvement from VOR grows. For the 10-task setting, OVOR-Deep can beat L2, which fine-tunes the entire pre-trained model, updating far more parameters than prompt-based methods (OnePrompt updates less 0.3% of total parameters).

As for results on CIFAR-100, CUB-200, and ImageNet-A. On these datasets, the prototype-based methods beat prompt-based methods, with the largest margin on CUB-200. Apart from that, One-Prompt generally performs better than OnePrompt-Deep on these datasets. Among the prompt-based methods, they exhibit similar ordering to that on ImageNet-R, and that VOR still always improves the accuracy for them.

The trend of the four prompt-based methods that is present in Table 2, indeed corroborates the design of OnePrompt. For CODA-P which beats OnePrompt, as it uses an attention layer to produce weights for combining all prompts in the pool, it resembles adding an additional transformer self-attention layer, explaining its edge over OnePrompt. Nonetheless, the edge of CODA-P does come with a cost: As indicated in Table 4, CODA-P has much more learnable

Table 4: Number of learnable parameters and FLOPs (and the ratio to OVOR) for the ImageNet-R 10-task setting.

| Method | FLOPs (G) | Learnable Parameters (M) |
|---|---|---|
| L2P | 35.18 (1.99×) | 0.64 (2.44×) |
| DualPrompt | 35.19 (1.99×) | 1.11 (4.26×) |
| CODA-P | 35.17 (1.99×) | 3.99 (13.28×) |
| OVOR | 17.60 (1.00×) | 0.26 (1.00×) |

parameters than other prompt-based methods. From Table 4 one can also see that OnePrompt only spends half inference cost as others, with only a fraction of number of learnable parameters.

Beside the main results in the paper, we perform ablation on following hyperparameters to validate our design choices: regularization weight $\lambda$, epochs used by regularized training, Huber or MSE loss in regularization loss, thresholds $\tau_{\text{Current}}, \tau_{\text{Outlier}}$ in the two hinge losses, and NPOS hyperparameters ($\sigma$, $\alpha$, $\beta$, and $K$), as well as where the prompts are inserted into the model. Their results are detailed in Appendix B, Tables B.7–B.16, respectively.

## 6 CONCLUSION

We introduce a novel way to regularize classifiers with synthetic outliers, for rehearsal-free class-incremental learning. Our method is shown to be compatible with different prompt-based methods, improving their performance across benchmarks. Nevertheless, currently OVOR is only applicable to scenarios where the input spaces of tasks are non-overlapping, that there is no obvious way to apply it to domain-incremental learning or blurred boundary continual learning (Buzzega et al., 2020). The outlier generation process also requires storing and searching among feature vectors, which may bring scalibility issues when the size of dataset grows. We are aware of the recent line of work which uses prototypes in lieu of the classifier head, and it would be an interesting future direction of our work to try different ways combining the two techniques. For instance, leveraging virtual outliers in calculating the prototypes, instead of just using class means. Finally, as OVOR is agnostic to the modality of the input, it would also be worthwhile to explore its performance on textual, tabular, or time-series data.

**Disclaimer** This paper was prepared for informational purposes by the Global Technology Applied Research center of JPMorgan Chase & Co. This paper is not a product of the Research Department of JPMorgan Chase & Co. or its affiliates. Neither JPMorgan Chase & Co. nor any of its affiliates makes any explicit or implied representation or warranty and none of them accept any liability in connection with this paper, including, without limitation, with respect to the completeness, accuracy, or reliability of the information contained herein and the potential legal, compliance, tax, or accounting effects thereof. This document is not intended as investment research or investment advice, or as a recommendation, offer, or solicitation for the purchase or sale of any security, financial instrument, financial product or service, or to be used in any way for evaluating the merits of participating in any transaction.

**Ethics statement.** We are not aware of potential ethic concerns regarding the proposed OVOR algorithm.

**Reproducibility statement.** Our efforts to facilitate reproducibility are summarized below:

- We describe the datasets we used in Section 5.
- We document the details and hyperparameters for reproducing prompt-based baselines in Section 5 and Section C.
- Our method is fully described in Section 4, with pseudo-code provided in Algorithm 1, and hyperparameters in Section 5 and Section C.

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

Table A.5: Results on ImageNet-R, for 5, 10 and 20 tasks.

| # of Tasks | 5 | | 10 | | 20 | |
|---|---|---|---|---|---|---|
| Method | $A_T$ | $F_T$ | $A_T$ | $F_T$ | $A_T$ | $F_T$ |
| L2 | – | – | 76.06±0.65 | – | – | – |
| A-Prompt | – | – | 72.82±0.30 | – | – | – |
| SimpleCIL | 61.28 | – | 61.28 | – | 61.28 | – |
| ADAM | 74.27 | – | 72.87 | – | 70.47 | – |
| L2P | 71.00±0.29 | 6.43±0.53 | 69.31±0.44 | 7.27±0.60 | 65.68±0.23 | 8.30±0.44 |
| L2P-VOR | 72.74±0.31 | 6.44±0.60 | 71.14±0.32 | 7.10±0.73 | 67.57±0.40 | 8.52±0.77 |
| DualPrompt | 73.11±0.21 | 5.66±0.42 | 71.47±0.48 | 6.02±0.45 | 67.91±0.32 | 6.51±0.32 |
| DualPrompt-VOR | 74.92±0.22 | 5.41±0.53 | 73.27±0.24 | 6.38±0.61 | 70.08±0.55 | 6.46±0.83 |
| CODA-P | 76.29±0.45 | 5.92±0.66 | 74.87±0.28 | 6.73±0.36 | 71.70±0.29 | 6.62±0.39 |
| CODA-P-VOR | 77.72±0.26 | 8.00±0.56 | 76.82±0.29 | 8.049±0.50 | 74.00±0.42 | 7.17±0.78 |
| OnePrompt | 74.65±0.48 | 4.60±0.31 | 73.34±0.71 | 5.36±0.53 | 70.42±0.08 | 5.57±0.25 |
| OnePrompt-Deep | 75.45±0.17 | 5.27±0.34 | 73.92±0.53 | 4.88±0.47 | 71.35±0.54 | 5.46±0.25 |
| OVOR | 76.79±0.31 | 4.88±0.69 | 75.61±0.45 | 5.77±0.53 | 73.13±0.35 | 6.06±0.78 |
| OVOR-Deep | 76.82±0.67 | 7.58±1.10 | 76.11±0.21 | 7.16±0.34 | 73.85±0.29 | 6.80±0.65 |

Table A.6: Results on CIFAR-100, CUB200, ImageNet-A

| Dataset | CIFAR-100 | | CUB200 | | ImageNet-A | |
|---|---|---|---|---|---|---|
| Method | $A_T$ | $F_T$ | $A_T$ | $F_T$ | $A_T$ | $F_T$ |
| A-Prompt | 87.87±0.24 | – | – | – | – | – |
| SimpleCIL | 76.21 | – | 85.16 | – | 49.44 | – |
| ADAM | 85.75 | – | 85.67 | – | 53.98 | – |
| L2P | 82.43±0.31 | 6.18±1.61 | 70.62±0.85 | 10.90±2.73 | 40.16±2.07 | 8.86±2.94 |
| L2P-VOR | 83.87±0.61 | 8.71±1.19 | 76.33±1.27 | 9.38±3.00 | 41.27±4.07 | 8.79±2.23 |
| DualPrompt | 83.28±0.60 | 5.75±1.08 | 71.85±1.14 | 10.31±2.34 | 44.85±2.39 | 10.80±1.73 |
| DualPrompt-VOR | 84.85±0.58 | 7.50±1.26 | 78.44±1.24 | 8.23±1.63 | 45.45±4.52 | 8.82±4.09 |
| CODA-P | 86.41±0.43 | 6.39±1.27 | 72.74±0.78 | 10.37±2.12 | 50.20±2.92 | 8.91±1.81 |
| CODA-P-VOR | 86.53±0.48 | 9.17±1.56 | 78.72±1.19 | 8.73±1.96 | 51.04±4.15 | 7.78±3.52 |
| OnePrompt | 85.60±0.41 | 4.79±1.25 | 72.19±0.88 | 10.18±2.30 | 47.36±2.79 | 9.75±1.80 |
| OnePrompt-Deep | 85.82±0.20 | 5.55±1.26 | 71.56±0.93 | 10.44±2.13 | 46.53±2.07 | 10.48±1.73 |
| OVOR | 86.68±0.26 | 5.25±1.38 | 78.12±1.14 | 8.73±1.87 | 48.49±4.30 | 7.83±3.92 |
| OVOR-Deep | 85.99±0.89 | 6.42±2.03 | 78.29±1.20 | 8.40±1.42 | 47.19±4.29 | 7.78±3.40 |

# APPENDIX

## A  COMPLETE EXPERIMENTAL RESULTS WITH STANDARD DEVIATIONS

## B  RESULTS FOR ABLATION STUDY

Here in this section we present the ablation results, for the options introduced by our regularization method. All experiments are conducted with OVOR on 10-task ImageNet-R setting, and we vary the hyperparameters around their default configuration detailed in Section 5.

Tables B.7–B.11 shows results for regularization weight $\lambda$, number of epochs for regularized training, the choice of $\mathcal{H}$ function in the definition in the regularization loss, and thresholds $\tau_{\text{Current}}$, $\tau_{\text{Outlier}}$, correspondingly. The following Tables B.12–B.15 shows those for NPOS hyperparameters $\sigma$, $\alpha$, $\beta$, and $K$. In these tables, the values in bold are the ones used by the default configuration. Most of them are rather insensitive to fluctuations, where the ones that can cause larger drops in accuracy are number of epochs and $\mathcal{H}$. However, note that even those with the lowest accuracy are still outperforming the one without the regularization. The number of epochs used by regularized training can affect the performance more significantly. If it is too large, the prompt will not have enough time to be fully-tuned, causing worse feature representations; if it is too small, the classifier head

---

**Algorithm 2** NPOS: Non-Parametric Outlier Synthesis

---

**Input:** Encoder $f_{\boldsymbol{\theta},\hat{\boldsymbol{p}}}$, Dataset $\mathcal{D}$, Distance measure $\Delta_K$, Hyperparameters $\alpha$, $\beta$, $\sigma$, $C$, $N$

$\mathcal{D}_{\text{feat}} \leftarrow \bigcup_{\boldsymbol{x}\in\mathcal{D}} f_{\boldsymbol{\theta},\hat{\boldsymbol{p}}}(\boldsymbol{x})$
$\mathcal{D}_{\text{boundary}} \leftarrow \text{topk}_{\boldsymbol{u}\in\mathcal{D}_{\text{feat}}}(\alpha \cdot C, \Delta_K(\boldsymbol{u}, \mathcal{D}_{\text{feat}}))$
$\mathcal{D}_{\text{noise}} \leftarrow \{\boldsymbol{n}_i \mid i \in \{1,\ldots,N\}, \boldsymbol{n}_i \sim \mathcal{N}(\boldsymbol{0}, \sigma\mathbf{I})\}$
$\mathcal{D}_{\text{candidate}} \leftarrow \{\boldsymbol{u} + \boldsymbol{n} \mid \boldsymbol{u} \in \mathcal{D}_{\text{boundary}}, \boldsymbol{n} \in \mathcal{D}_{\text{noise}}\}$
$\mathcal{D}_{\text{outlier}} \leftarrow \text{topk}_{\boldsymbol{v}\in\mathcal{D}_{\text{candidate}}}(\beta \cdot C, \Delta_K(\boldsymbol{v}, \mathcal{D}_{\text{feat}}))$
**return** $\mathcal{D}_{\text{outlier}}$

---

will be under-regulated. For function $\mathcal{H}$, the difference between Huber loss and MSE loss is how they respond to values far away from the target. Due to how the outlier generation algorithm NPOS works, it is possible for it to output samples that actually lie within the training samples. When such samples are used as outliers, they generate energy scores very far from the outlier threshold $\tau_{\text{Outlier}}$. In this case, MSE will generate much larger loss value than Huber loss, as it squares the distance in its calculation, being more sensitive to such errors.

### B.1 ABLATION ON POSITION OF INSERTING PROMPT

As mentioned in Section 5, by default OnePrompt inserts length-5 prompts in first two layers, and length-20 prompts in subsequent three layers. Here we report the results varying the insertion positions, in Table B.16. The first row is the configuration for OnePrompt. As can be seen in the table, this configuration is not the best, but we use it to match the configuration of DualPrompt, in order to make direct comparison. The second and third rows insert same-length prompts to all layers, which is same as VPT-Deep, achieve best accuracy among all. We choose the second one, with length 5, for OnePrompt-Deep, because it uses less prompt parameters than OnePrompt.

## C IMPLEMENTATION DETAILS

We build our implementation upon the official code repository of CODA-P (Smith et al., 2023a) [2], which also implements two other prompt-based methods: L2P (Wang et al., 2022c) and DualPrompt (Wang et al., 2022b). We make the several modifications on it, implementing OnePrompt and VOR, and we added code for better reproducibility when building dataloaders in parallel.

For all prompt-based methods, their pre-trained ViT-B checkpoints are provided by the timm library [3], which in turn use the one trained by Steiner et al. (2022) [4]. Please see Table C.17 for complete configuration.

## D ALGORITHM FOR GENERATING VIRTUAL OUTLIERS

Algorithm 2 shows the NPOS algorithm used for generating virtual outliers, with $C$ being number of classes per task, and $N$ a sufficiently large integer (we used 600 as in (Tao et al., 2023)). NPOS defines distance measure $\Delta_K(\boldsymbol{x}, \mathcal{D})$ as the distance between the point $\boldsymbol{x}$ and its K-nearest-neighbor in $\mathcal{D}$, while $\text{topk}_{\boldsymbol{x}\in\mathcal{D}}(K, \cdot)$ works similarly to $\arg\max_{\boldsymbol{x}\in\mathcal{D}}(\cdot)$, but instead returns top $K$ elements.

---

[2] https://github.com/GT-RIPL/CODA-Prompt
[3] https://github.com/pprp/timm
[4] https://storage.googleapis.com/vit_models/augreg/B_16-i21k-300ep-lr_0.001-aug_medium1-wd_0.1-do_0.0-sd_0.0--imagenet2012-steps_20k-lr_0.01-res_224.npz

Table B.7: Ablation result on weight $\lambda$ for regularization loss.

| $\lambda$ | 0.05 | **0.1** | 0.5 | 1.0 |
|---|---|---|---|---|
| $A_T$ | 75.23 | 75.61 | 75.61 | 75.58 |

Table B.8: Ablation result on number of epochs using regularized-training.

| Last (%) of total epochs | 10% | **20%** | 30% |
|---|---|---|---|
| $A_T$ | 74.42 | 75.61 | 73.99 |

Table B.9: Ablation result on function $\mathcal{H}$ in regularization loss calculation.

| $\mathcal{H}$ | **Huber** | MSE |
|---|---|---|
| $A_T$ | 75.61 | 73.75 |

# E    ADDITIONAL ANALYSES ON REPRESENTATION DRIFT

In order to prove that OnePrompt does not suffer from representation drift even when its prompt is updated throughout all tasks, we calculate the drift in its prompt and the representation vectors of same inputs. Table E.19 shows the cosine similarity of the prompt parameters concatenated between consecutive tasks, e.g. prompt parameters before and after training on task 2 data have cosine similarity of 99.26%. According to it, change in prompt parameters of OnePrompt is marginal, and can explain why OnePrompt has minor representation drift. This conclusion is further reinforced by the results in Table E.20, where we measure the similarity in representation of test samples, in a similar way of Table E.19. In addition to cosine similarity, we include the CKA similarity metrics (Kornblith et al., 2019) as well.

Table B.10: Ablation result on energy score threshold $\tau_{\text{Current}}$ for training data samples.

| $\tau_{\text{Current}}$ | -21.0 | **-24.0** | -27.0 |
|---|---|---|---|
| $A_T$ | 75.61 | 75.61 | 75.57 |

Table B.11: Ablation result on energy score threshold $\tau_{\text{Outlier}}$ for outliers.

| $\tau_{\text{Outlier}}$ | 0.0 | **-3.0** | -6.0 |
|---|---|---|---|
| $A_T$ | 75.38 | 75.61 | 75.70 |

Table B.12: Ablation result on covariance parameter $\sigma$ in generating outliers.

| $\sigma$ | 0.1 | 0.5 | **1.0** | 2.0 |
|---|---|---|---|---|
| $A_T$ | 75.57 | 75.57 | 75.61 | 75.59 |

Table B.13: Ablation result on $\alpha$, which is used to calculate number of selected boundary samples.

| $\alpha$ | 5 | 7.5 | **10** | 12.5 | 15 |
|---|---|---|---|---|---|
| $A_T$ | 75.60 | 75.58 | 75.61 | 74.58 | 74.46 |

Table B.14: Ablation result on $\beta$, which is used to calculate number of generated outliers.

| $\beta$ | 100 | 120 | 140 | **160** | 180 | 200 |
|---|---|---|---|---|---|---|
| $A_T$ | 75.48 | 75.50 | 75.58 | 75.61 | 75.58 | 74.60 |

Table B.15: Ablation result on $K$ in KNN distance, used in outlier generation.

| $K$ | 50 | **100** | 150 |
|---|---|---|---|
| $A_T$ | 75.38 | 75.61 | 75.38 |

Table B.16: Ablation result on prompt position, with OnePrompt on 10-task ImageNet-R. The left-most column is the start and end layer for length-5 prompts, and the middle column for length-20.

| Length-5 | Length-20 | Accuracy |
|---|---|---|
| (1,2) | (3,5) | 73.34 |
| (1,12) | − | 73.92 |
| − | (1,12) | 74.69 |
| (3,4) | (3,5) | 73.46 |
| (5,6) | (3,5) | 73.41 |
| (6,7) | (3,5) | 73.89 |
| (1,2) | (2,4) | 72.89 |
| (1,2) | (4,6) | 73.55 |
| (1,2) | (5,7) | 73.93 |

Table C.17: Detailed configuration for implementation.

| | |
|---|---|
| Random seeds | 0, 1, 2, 3, 4 |
| Input image size | $(3, 224, 224)$ |
| Input normalization | mean $(0, 0, 0)$, std $(1, 1, 1)$ |
| Epochs | 50 on ImageNet-R, 20 on CIFAR-100 |
| Batch size | 128 |
| Optimizer | Adam, $\beta_1 = 0.9$, $\beta_2 = 0.999$ |
| Learning rate | $1e^{-3}$ |
| Schedule | Cosine decay |
| Number of epochs for VOR | Last 20% of total epochs |
| Huber loss | $\delta = 1.0$ |
| Image Preprocessing | |
| ImageNet-R (train) | RandomResizedCrop(224), RandomHorizontalFlip(0.5), Normalize() |
| ImageNet-R (test) | Resize(256), CenterCrop(224), Normalize() |
| Other (train) | RandomResizedCrop(224), RandomHorizontalFlip(0.5), Normalize() |
| Other (test) | Resize(224), Normalize() |
| L2P | Prompt length: $(5, 5, 5, 5, 5)$, Prompt pool size: 30 |
| DualPrompt | Prompt length: $(5, 5, 20, 20, 20)$, Prompt pool size: one per task |
| CODA-P | Prompt length: $(4, 4, 4, 4, 4)$, Prompt pool size: 100 |
| OnePrompt | Prompt length: $(5, 5, 5, 20, 20)$, Prompt pool size: 1 |
| OnePrompt-Deep | Prompt length: $(5, 5, 5, 5, 5)$, Prompt pool size: 1 |
| Prompt insertion method | Prefix-Tuning (Li & Liang, 2021) |

Table C.18: VOR Hyperparameters for each dataset.

| Dataset | $\lambda$ | $\tau_{\text{Current}}$ | $\tau_{\text{Outlier}}$ | $\sigma$ | $\alpha$ | $\beta$ | $K$ |
|---|---|---|---|---|---|---|---|
| ImageNet-R | 0.1 | -24.0 | -3.0 | 1.0 | 10 | 160 | 100 |
| CIFAR-100 | 0.1 | -24.0 | -3.0 | 1.0 | 10 | 160 | 100 |
| CUB-200 | 0.1 | -24.0 | -3.0 | 0.1 | 4 | 60 | 100 |
| ImageNet-A | 0.1 | -15.0 | -3.0 | 1.0 | 4 | 60 | 100 |

Table E.19: Drift in Prompt Parameters. For OnePrompt on 10-task ImageNet-R, we calculate the cosine similarity between the prompt before and after training of each task.

| Task 2 | Task 3 | Task 4 | Task 5 | Task 6 | Task 7 | Task 8 | Task 9 | Task 10 |
|---|---|---|---|---|---|---|---|---|
| 99.26% | 99.60% | 99.53% | 99.78% | 99.81% | 99.87% | 99.82% | 99.88% | 99.89% |

Table E.20: Drift in feature representations. For OnePrompt on 10-task ImageNet-R, we calculate the cosine and CKA similarity between the representations of test samples of a task before and after training that task.

| Metric | Task 2 | Task 3 | Task 4 | Task 5 | Task 6 | Task 7 | Task 8 | Task 9 | Task 10 |
|---|---|---|---|---|---|---|---|---|---|
| Cosine | 88.28% | 88.69% | 88.21% | 89.30% | 89.18% | 89.02% | 88.79% | 88.78% | 89.00% |
| CKA (linear) | 91.63% | 92.70% | 90.59% | 93.10% | 93.39% | 92.72% | 92.34% | 92.68% | 92.53% |
| CKA (kernel) | 92.12% | 93.22% | 91.13% | 93.65% | 93.92% | 93.36% | 92.93% | 93.25% | 93.25% |

