# OpenReview forum: "OVOR: OnePrompt with Virtual Outlier Regularization for Rehearsal-Free Class-Incremental Learning"
_ICLR.cc/2024/Conference — ICLR 2024 poster_

### Official Review · Reviewer_DjrG · 2023-10-30

**Soundness:** 2 fair
**Presentation:** 3 good
**Contribution:** 3 good
**Rating:** 6
**Confidence:** 5

**Summary:**

This paper describes a method for class incremental learning (CIL) that makes use of frozen pre-trained ViT transformer networks. The essence of the method is to learn prompts, while keeping the pre-trained network frozen, and hence ensure it is unable to forget during CIL. Past work in this area, namely L2P, DualPrompt and CODA-P, is therefore highly relevant.  The paper identifies several weakness with those past methods for prompting ViT networks for CIL, and proposes improvements to the general methodology. Firstly, it proposes the use of "VOR", i.e. Virtual Outlier Regularization, in which the learned output head is trained further following freezing of newly learned prompts, on a combination of feature vectors, and synthesised outliers. The idea here is to try to tighten the decision boundaries between classes. Secondly, it proposes OnePrompt, a simplification to past prompting mechanisms, namely removing the use of different prompt sets for different tasks. When VOR and OnePrompt are combined (as OVOR), the paper shows better results than L2P and DualPrompt, with comparable or better results than CODA-P.  The number of parameters and FLOPs is reduced in OVOR compared to those methods.

**Strengths:**

- Originality: the paper isolates weaknesses in comparison methods and proposes a solution, namely the VOR method, which is the first use of such a method in the prompting of ViT models for CIL. Moreover, the proposed simplification to past prompting methods for CIL is novel, as is the combination of VOR and OnePrompt
- Quality: the paper sets out the context and problem setting well, clearly explains the rationale for the methods, and tests the method on datasets that are of clear interest in the CIL community.
- Clarity: the writing is clear and the structure easy to follow; this is why I have very few questions.
- Significance: Given prompting methods using pre-trained ViTs are very new in CIL, this is a potentially important paper. I think the community is currently unclear on whether prompting methods can be significantly improved on, and there is a need for studies such as this in which simplifications, parameter efficiency and additional regularization are used to enhance prompting methods.

**Weaknesses:**

- 1. The main weakness of the paper is that important comparison results in the use of pretrained ViT networks are omitted. The significance of the contribution in this paper would be much easier to appraise if these are added by the authors.
    - 1.1 First, there are two omitted relevant papers that use alternative methods to prompting. These both produce higher final average accuracies using pre-trained ViT models than any reported here.  The first of these is the SLCA method [1]. The second is the RanPAC method [3] that builds on the work of the paper by Zhou et al [2], which is already cited in the paper.
        - In the current paper, OVOR on ImageNet-R achieves 75.6% for T=10, and CIFAR-100 achieves 86.7% for T=10.
        -  In ref [1], ImageNet-R achieves 77% for T=10 and CIFAR-100 achieves 91.53% for T=10.
        - In ref [3], ImageNet-R achieves 78.1% for T=10 and for CIFAR100 achieves 92.2% for T=10 ( T=5 and T=20 results can also be seen in Table A4 of [3]).
    - 1.2 Second, a clear distinction between refs [1,2,3] and the current paper is that [1,2,3] report results on many more datasets, whereas here only two are used: split CIFAR-100 and split ImageNet-R.
        - The current paper's contribution and standing relative to refs [1,2,3] would be much clearer if results for OVOR are reported on at least 4 datasets. E.g. [1] also reports results on split CUB and split Cars, while [2] and [3] each report on 7 datasets for CIL in total. There is no good reason to not demonstrate OVOR's performance on at least 4, and preferable 6 datasets such as those in [1,2,3].  This is particularly important given that table 2 shows that OnePrompt alone produces lower final average accuracy than CODA-P, and only the combination of OnePrompt with VOR does better than CODA-P.
    - Note: I am aware that paper [3] (and possibly [1]) might be deemed "contemporaneous" according to the FAQ on this point (which is unclear re arXiv). Therefore, my recommendations to compare to those are aimed at improving the value of the paper should it be published. However, my recommendation to add more datasets is independent of any date comparisons, especially given Zhou et al's other datasets could have been used to add more dataset comparisons, to help the reader understand whether there truly is a trend for the method to outperform coda-prompt, DualPrompt etc.

- 2. The paper does not thoroughly explore  ablation results with regards to the combination of OnePrompt or other prompting methods with VOR.  While ablations for OnePrompt are shown for ImageNet-R in Table 2, there is no data for CIFAR-100.  My above comments re more datasets extends to this point - for the reader to understand the contributions of OnePrompt and VOR both individually and combined, we really need to see data for ablations on more than just a single dataset. Moreover, the authors state "The proposed virtual outlier regularization (VOR) is agnostic to any prompt-based CIL." This therefore means it is of high interest to know how well VOR combines with CODA-P, given that CODA-P without VOR outperforms OnePrompt without VOR.

- 3. On page 3, in the second paragraph, the authors refer to the prototype methods of Zhou et al and Ma et al and state "However, it is unclear whether they can be considered rehearsal-free, as they claim to be."
    - This is a misleading statement that should be deleted. Rehearsal methods require storage of samples from past tasks for inclusion in training in newer tasks.  In my view it's very well understood in the CL community that this means retaining a copy of the full original datasample, e.g. an input image, and using such samples individually alongside new samples to update the parameters of a model. This is really quite different to class prototypes as used by Zhou et al (and also, for example, Ref [3]). In Zhou et al, class prototypes are firstly certainly not the original samples, as they are formed from the output feature representations (e.g. a length 512 vector instead of an RGB image). More importantly, prototypes are an average over all the length 512 vectors for a class, so the result cannot in any way be thought of in the same way as "rehearsal", since there is no way to recover individual samples, and in this sense are more like the weights in a neural network than actual samples. Finally, and most important of all for this point, in Zhou et al  class prototypes from past tasks are not used to update any parameters in any subsequent task, which is in total contrast to the entire point of  rehearsal memory. In summary, I think its false to suggest that Zhou et al and related methods are not "rehearsal free".
    - The authors then make a statement about privacy concerns - this is a totally independent point to the distinction between class prototypes and rehearsal memory, but I am skeptical about how much privacy concern there can be about a class-mean vector. I'm willing to stand corrected, but I cannot see any chance of original samples being extracted from a mean created from many orders of magnitude of samples to form a feature vector....
    - The inclusion of these statements has the appearance of being an attempt to justify why a prompt based CIL method should be preferred over a prototype method like Zhou et al. But I don't think it is necessary to try to justify this. Regardless of whether prototype methods may or may not achieve better results than prompting methods, the current paper's advantages over past prompting methods are still potentially important, because (i) there may be benefits from combining prompting and prototype methods, which to my knowledge has not yet been done; and (ii) it could happen that future improvements on ViT transformer networks may benefit more from prompting than they do from prototype methods.

- 5. Minor point: the paper's intro assumes the reader understands well that the approaches in the paper apply only to pre-trained ViT networks. For the larger Continual Learning community, this is still unusual as most methods for CL focus on training CNNs such as ResNets, and on mitigating forgetting when the benefits of a pre-trained model are not available. The paper would benefit from making these points clear early on.

- Summary: I have chosen a rating of 5 at this time. In order to reconsider my rating, the authors will need to (i) provide the reader with a clearer view on how OVOR compares to other methods on more datasets  (and, ideally, comparison with refs [1,3] as well), and (ii) show thorough ablations compared with the best competing prompting method, CODA-P, i.e. CODA-P with and without VOR, and OnePrompt with and without VOR, all on at least 4 datasets.

- Refs:
    - [1] SLCA: https://arxiv.org/abs/2303.05118 (submitted to arXiv 9 March 2023, now published in ICCV 2023).
    - [2] https://arxiv.org/abs/2303.07338 (cited in the paper as Zhou et al. (2023), submitted to arXiv 13 March 2023)
    - [3] RANPAC: https://arxiv.org/abs/2307.02251 (submitted to arXiv 5 July 2023, now accepted in NeurIPS 2023).

**Questions:**

This is repeating the content of my response under "Weaknesses": my questions are these:

1. What is the performance of VOR when combined with L2P, DualPrompt and CodaPrompt, and how does that compare with OVOR?
2. What is the performance of OVOR, and ablations (VOR alone and OnePrompt alone) on at least 2 more datasets?

---

> ### Author Response · Authors · 2023-11-23
> **Response to Reviewer DjrG - W1, W2**
>
> We are grateful of the reviewer's valuable comments and insights, and we are very delighted that you find our work original and important.
>
> **W1.1 First, there are two omitted relevant papers that use alternative methods to prompting.**
>
> Thank you for pointing it out those relevant and contemporaneous works, we add the discussion of those two papers [Zhang et al., 2023][McDonnell et al., 2023] in the related works section.
> Moreover, we would like to point out one experimental setting in their papers, even though they are using ViT as the backbone but they are actually using different pre-trained checkpoints from ours. Thus, it makes harder to directly compare the performance to their results.
>
> Given the time of rebuttal, it is beyond our computing resource to re-run all experiments based on another checkpoint for ViT given that we have conducted the suggested experiments on more datasets and more ablation studies.
> Nonetheless, we think how to combine the virtual outliers to prototype-based method will be an interesting research direction.
>
> References:
>
> - Gengwei Zhang, Liyuan Wang, Guoliang Kang, Ling Chen, Yunchao Wei, SLCA: Slow Learner with Classifier Alignment for Continual Learning on a Pre-trained Model, ICCV 2023
> - Mark D. McDonnell, Dong Gong, Amin Parveneh, Ehsan Abbasnejad, Anton van den Hengel. RanPAC: Random Projections and Pre-trained Models for Continual Learning, NeurIPS 2023.
>
> **W1.2 The current paper's contribution and standing relative to refs would be much clearer if results for OVOR are reported on at least 4 datasets.**
>
> **W2. The paper does not thoroughly explore ablation results with regards to the combination of OnePrompt or other prompting methods with VOR.**
>
> Thanks for the suggestion, we address these two points together (and two questions). We further compare the performance of VOR over the prompt-based method, including L2P, DualPrompt, CODA-P and OnePrompt on ImageNet-R, CIFAR-100, CUB-200 and ImageNet-A, and the results are shown in the following table. The proposed VOR consistently improve all prompt-based methods on all datasets, which shows the VOR could be used in many types of datasets. We updated the results in Table 2 of the revision.
>
> **Table**: The CIL accuracy on four datasets under the 10-task setting. The number in the parenthesis denotes the results with VOR.
>
> | Method | ImageNet-R | CIFAR-100 | CUB-200 | ImageNet-A |
> | --- | --- | --- | --- | --- |
> | L2P | 69.31 (71.14) |  82.43 (83.87) | 70.62 (76.33) | 40.16 (41.27) |
> | DualPrompt | 71.47 (73.27) | 83.28 (84.85) | 71.85 (78.44) | 44.85 (45.45) |
> | CODA-P | 74.87 (76.82) | 86.41 (86.53) | 72.74 (78.72) | 50.20 (51.04) |
> | OnePrompt | 73.34 (75.61) | 85.60 (86.68) | 72.19 (78.12) | 47.36 (48.49) |

---

> > ### Author Response · Authors · 2023-11-23
> > **Response to Reviewer DjrG - W3, W5**
> >
> > **W3. On page 3, in the second paragraph, the authors refer to the prototype methods...**
> >
> > Thank you for the remarks, we removed the pertinent two sentences in the revision, because it seems that they are giving the impression that we do not intend to (to justify that prompt-based methods are preferred over prototype-based methods).
> > Nonetheless, we would like to discuss more thoroughly about certain parts in the reviewer's comments.
> >
> > First of all, the reviewer mentioned that rehearsal means "retaining copy of full original data sample", but for works that learn a generative model from past data [Ostapenko et al., 2019][van der Van et al., 2020], a.k.a. pseudo-replay or deep generative replay, they are still considered as rehearsal-based methods [Smith et al., 2023a][Mai et al., 2022].
> > Of course, this by no means imply that prototype-based methods are not rehearsal-free, and it is also true that the prototypes are not used along with new inputs to update the model.
> > To us it is more like a spectrum, depending on how much information about past data is stored, with original replay-based on one end and having only classifier weights on the other, and between them there are pseudo-replay (storing generative models), SLCA (storing class distributions), prototype-based (storing class means).
> > We updated the discussion of prototype-based method in the related works.
> >
> > Next, regarding the privacy concerns, we totally agree that class-means contain very little information compared to replay buffers.
> > The original samples cannot be extracted from class means alone, but that may not be the case when we have class means along with the encoder weights.
> > In fact, there exists work [Haim et al., 2022] that tries to reconstruct original data from the classifier, though it is done on simpler networks and smaller datasets, hence there is a long way for this to become a pressing issue for networks and datasets as large and complex as we are using now.
> > Nonetheless, it demonstrates the possibility of extracting training samples, even without prototypes.
> > And certainly storing prototypes would provide extra information on top of that.
> > Hence in our view, similar as aforementioned, there is a spectrum with these methods, where storing prototypes provides more information about past data than without, which makes it easier to achieve better CL performance, but also for reconstruction of past data samples.
> >
> > Reference:
> >
> > - Oleksiy Ostapenko, Mihai Marian Puscas, Tassilo Klein, Patrick J ̈ahnichen, and Moin Nabi.
> > Learning to remember: A synaptic plasticity driven framework for continual learning. In
> > IEEE Conference on Computer Vision and Pattern Recognition, CVPR 2019, Long Beach, CA,
> > USA, June 16-20, 2019, pp. 11321–11329.
> > - Gido M. van de Ven, Hava T. Siegelmann, and Andreas Savas Tolias. Brain-inspired replay for
> > continual learning with artificial neural networks. Nature Communications, 11, 2020.
> > - James Seale Smith, Leonid Karlinsky, Vyshnavi Gutta, Paola Cascante-Bonilla, Donghyun Kim,
> > Assaf Arbelle, Rameswar Panda, Rogerio Feris, and Zsolt Kira. Coda-prompt: Continual de-
> > composed attention-based prompting for rehearsal-free continual learning. In Proceedings of the
> > IEEE/CVF Conference on Computer Vision and Pattern Recognition (CVPR), pp. 11909–11919,
> > June 2023a.
> > - Zheda Mai, Ruiwen Li, Jihwan Jeong, David Quispe, Hyunwoo Kim, and Scott Sanner. On-
> > line continual learning in image classification: An empirical survey. Neurocomputing, 469:
> > 28–51, 2022.
> > - Niv Haim, Gal Vardi, Gilad Yehudai, Ohad Shamir, Michal Irani. Reconstructing Training Data from Trained Neural Networks, NeurIPS 2022
> >
> >
> >
> > **W5. The paper’s intro assumes the reader understands well that the
> > approaches in the paper apply only to pre-trained ViT networks. For the larger
> > Continual Learning community, this is still unusual as most methods for CL
> > focus on training CNNs such as ResNets, and on mitigating forgetting when the
> > benefits of a pre-trained model are not available.**
> >
> > Thanks for the suggestion, we have made changes in Section 2 and 3.2 when CODA-Prompt, DualPrompt and L2P are mentioned.
> > Nonetheless, we would like to clarify that VOR can also possibly be applied to networks other than ViT.
> > As it uses features produced by the encoder to regularize the classifier head, there is no dependency on network architecture of the encoder, or whether it is pre-trained.

---

### Official Review · Reviewer_Dx4f · 2023-10-30

**Soundness:** 2 fair
**Presentation:** 1 poor
**Contribution:** 3 good
**Rating:** 6
**Confidence:** 4

**Summary:**

There are two primary motivations in this paper. The first one is to show that the need to have a pool of prompts (something that most prompt-based approaches in CL do) is not necessary. For this, the authors propose OnePrompt, which is a simple baseline that uses only one prompt to learn all tasks instead of having task-specific prompts. The second motivation is that rehearsal-free approaches struggle to differentiate classes from different tasks since they are never optimized simultaneously. For this, the authors propose regularizing the decision boundary using  Virtual Outliers. The authors mentioned that these outliers help to define the decision boundary of each class better. The results show promising results in two commonly used benchmarks. Surprisingly, OnePrompt does not present high forgetting.

**Strengths:**

- The idea of having a OnePrompt method is interesting. And it surprised me how well it worked.
    - I especially like the idea of reducing the computational cost by not requiring to find the closest key-pair values.
- The results showed that the proposed regularization method helps improve the accuracy. The motivation is clear.
- The paper shows promising results in 2 standard benchmarks. However, only state-of-the-art results in one.
- The background section in a paper is very helpful since it presents the notation and everything it needs to know about the contributions

**Weaknesses:**

- I understand and share the motivation that rehearsal-free approaches struggle to distinguish classes from different tasks. However, it would be good to add a more realistic example, values or theory that support this statement.
    - Or at least something that shows what you think is happening.
- In multiple parts of the paper, it is mentioned that prompt-based approaches require a pool of task-specific prompts. Although I understand the idea behind it, this is only partially true, as the authors say in section 3.2. They explain that the selection depends on a function; ideally, one would expect this feature to be able to be separated by task, but it is not a requirement of the methods.
- The structure of the paper needs to be clarified. For example, why present Virtual Outlier Regularization first rather than OnePrompt?
    - At least for me, OnePrompt is like a simple baseline. It's a good one, as it shows in the results, but only a base method to be compared.
    - I believe that adding Virtual Outlier to other methods is the main contribution. However, how to obtain the virtual outliers is not explained in the paper. I understand that citing is a proper way of explaining it. However, as it has been such an essential part of the contribution, it is better to have a small explanation.
- In Table 3, it should be good to include OnePrompt for comparison. Also, add some vertical lines to separate # of tasks clearly.

**Questions:**

- It is not clear which setting OnePrompt works. Does it work in a Class incremental or task incremental setting (with a share classifier)?
    - This confusion appears especially after seeing Table 1. Why is this table important? How does the same experiment behave in CIL?
- Is there an intuition on how and why OnePrompt works so well?
    - How much does the prompt change in each task? Since there is no forgetting, it shouldn't change much or no?
    - How does the representation of previous tasks change after training new tasks?
- Have you run experiments when training only the classifier? Together with using the approach mentioned in Eq 3.
- Is there an explanation for why the VOR increases forgetting? As shown in Table 2.
- In Section 5, you mentioned that for OnePrompt, you insert the prompts into the first 5 layers. Did you do an ablation of this hyperparameter?
- How costly is the VOR approach?
- Do you have the results for OnePrompt for Table 4? In the last 2 paragraphs, you mentioned a comparison between OnePrompt and other methods, however, I don't see it in the tables. Are you referring to OVOR?

---

> ### Author Response · Authors · 2023-11-23
> **Response to Reviewer Dx4f - W1, W2, W3**
>
> We appreciate the remarks and observations the reviewer has shared, and thank the reviewer that recognizing our contribution in reducing the computational cost
> and considering our literature review helpful. Here we addressed the reviewer's concerns point-by-point.
>
> **W1. I understand and share the motivation that rehearsal-free approaches struggle to distinguish classes from different tasks. However, it would be good to add a more realistic example, values or theory that support this statement.**
>
> Thank you for pointing out the confusion, let us clarify it.
> First of all, there are two types of misclassification in the CIL setting, the model misclassifies the data: (I) as another class in *the same task* or (II) as a class from *a different task*.
> The rehearsal-free method suffers the second type of misclassification mostly since the there is no data from other tasks can be used to help the model differentiate data from different tasks.
>
> To provide better illustration why we should be more concerned about the latter one, we evaluated OnePrompt (can be any prompt-based method) under TIL (task ID is given during evaluation) and CIL settings on 10-task ImageNet-R, and the results are shown in the following table.
> Note that the key difference between TIL and CIL is that the task ID is given in TIL, such that the model won't have any misclassifications for the second type.
> The numbers in the table denote the performance change between the first (right after each new task) and last (task 10) evaluation on each task.
> Under the TIL setting, OnePrompt retains its performance on each task; this suggests that there is negligible influence from the first type of misclassification (i.e., misclassifiying the data as another class from the same task).
> On the other hand, the performance drops significantly under the CIL setting; thus, it brings us to the conclusion that performance degradation majorly comes from the second type of misclassification (misclassifiying the data as another class in a different task).
>
> **Table**: Performance change of OnePrompt when evaluated under TIL and CIL on 10-task ImageNet-R. The changes are measured between the first (right after each task) and last evaluation (task 10).
>
> | Type | Task 1 | Task 2 | Task 3 | Task 4 | Task 5 | Task 6 | Task 7 | Task 8 | Task 9 | Average |
> |---|---|---|---|---|---|---|---|---|---|---|
> |TIL|+2.05|+1.64|+1.34|+0.40|+0.61|+0.24|+0.23|-0.04|+0.36|+0.76|
> | CIL | -16.24 | -7.53 | -7.22 | -5.05 | -5.31 | -4.19 | -2.84 | -1.45 | -1.44 | -5.36 |
>
> **W2. In multiple parts of the paper, it is mentioned that prompt-based approaches require a pool of task-specific prompts.**
>
> It is true that the prompt selection can be non-task-specific, e.g. CODA-Prompt uses an attention mechanism to obtain a weighted combination of all prompts.
> Nonetheless, the learning of those prompts is not. Again in CODA-Prompt, the prompt pool divided into a number of partitions, and each partition will only be updated by data from one task, and frozen afterwards.
> To say in very high level way, what benefit having multiple prompts brings is the ability to isolate knowledge from different tasks, otherwise it can viewed as a single prompt updated continuously, just with more parameters.
> This is why we refer them as "task-specifc" prompt pool.
>
> **W3. The structure of the paper needs to be clarified. For example, why present Virtual Outlier Regularization first rather than OnePrompt?**
>
> The inter-task confusion is the most important issue in all rehearsal-free methods, and the proposed VOR is a general solution for the prompt-based methods (as shown in Table 2); thus, the VOR is our main contribution, and then we introduce it first.
> After having VOR to help different prompt-based methods, we further explore different prompt-based methods instead of directly adopting their prompt-based methods, and we find that the simple OnePrompt achieved superior performance.
> With this regard, we believe that it is better if OnePrompt is not presented before VOR, breaking the aforementioned narrative.
>
> **W3-2. I believe that adding Virtual Outlier to other methods is the main contribution...**
> Thank you for mentioning this, we have included add the description and the pseudo codes in the revision, see Appendix D and Algorithm 2.

---

> ### Author Response · Authors · 2023-11-23
> **Response to Reviewer Dx4f - W4, Q1, Q2**
>
> **W4. In Table 3, it should be good to include OnePrompt for comparison. Also, add some vertical lines to separate number of tasks clearly.**
>
> Thank you pointing it out, we have added the row of OnePrompt in Table 3 of the revision .
>
> **Q1. It is not clear which setting OnePrompt works.**
>
> OnePrompt works in both TIL and CIL setting (with a shared classifier), but the majority part of our work, along with the experiments, is investigating and addressing problems specific to the CIL setting.
> In our paper, we also mentioned that OnePrompt is better than L2P and DualPrompt in CIL setting (Section 4.2 and Table 2).
> We have addressed the confusion in the revision, that the purpose of having the results in Table 1 is to provide the insight that OnePrompt only has minor representation drift even with its prompt continuously updated by all tasks.
>
> **Q2. Is there an intuition on how and why OnePrompt works so well?**
>
> We believe it is more appropriate to explain the other way around: why DualPrompt has the problem in obtaining better results than OnePrompt.
> In the design of DualPrompt, it tries to build an 1-to-1 correspondence between prompts and tasks.
> Selecting the correct prompt ideally should be important for high classification accuracy, otherwise it would contradict with the intention of the design.
> Nonetheless, from the way DualPrompt updates the prompt keys, it is obvious that the keys will converge to the mean of feature vectors of one task, which does not provide too much semantics as the class composition of a task is fully random.
> So in inference, each input is just matched to the closest task mean, hence we do not expect it to have high test matching accuracy.
> In fact, in the DualPrompt paper, they reported the prompt matching accuracy of **55.8\%**.
> If the prompts are not selected correctly, then it would not help with the classification further down the road.
> Therefore, it would be limited how much the task-specific prompt mechanism of DualPrompt can help the classification accuracy.
>
> **Q2-1 How much does the prompt change in each task?**
>
> Thanks for the insightful question. To address that, we conducted the following analysis on OnePrompt to measure the prompt changes after each task on 10-task ImageNet-R. The following analysis could further provide the evidence that the drift of feature representation is marginal for OnePrompt.
> We compared the concatenation of all prompt parameters after training for the first and last task, and the cosine similarity between two consecutive tasks are shown in the table below.
> The prompt changes are marginal in each task and this result agrees with our observation on the feature drift of OnePrompt. We will add this analysis into our revision to provide more insights why the OnePrompt has minor representation drift. (Table E.19)
>
> **Table**: Prompt similarity between two consecutive tasks for OnePrompt on 10-task ImageNet-R.
>
> | Task 2 | Task 3 | Task 4 | Task 5 | Task 6 | Task 7 | Task 8 | Task 9 | Task 10 |
> |---|---|---|---|---|---|---|---|---|
> | 99.26\% | 99.60\% | 99.53\% | 99.78\% | 99.81\% | 99.87\% | 99.82\% | 99.88\% | 99.89\% |
>
> **Q2-2. How does the representation of previous tasks change after training new tasks?**
>
> Thanks to point out this suggestion to further understand the representation drift in OnePrompt.
> We compared the similarity of feature representations for all test samples of ImageNet-R, produced by models trained before and after one task, and the results are shown in following table.
> We measure the similarity based on cosine similarity, Centered Kernel Alignment (CKA) [Kornblith et al., 2019] with linear and kernel variants.
>
> **Table**: The representation similarity between two consecutive tasks for OnePrompt on 10-task ImageNet-R.
>
> | Task 2 | Task 3 | Task 4 | Task 5 | Task 6 | Task 7 | Task 8 | Task 9 | Task 10 |
> |---|---|---|---|---|---|---|---|---|
> |Cosine Similarity| 88.28\% | 88.69\% | 88.21\% | 89.30\% | 89.18\% | 89.02\% | 88.79\% | 88.78\% | 89.00\% |
> |CKA (linear)|91.63\%|92.70\%|90.59\%|93.10\%|93.39\%|92.72\%|92.34\%|92.68\%|92.53\%|
> |CKA (kernel)|92.12\%|93.22\%|91.13\%|93.65\%|93.92\%|93.36\%|92.93\%|93.25\%|93.25\%|
>
> Reference:
> - Kornblith, Simon, Mohammad Norouzi, Honglak Lee, and Geoffrey Hinton. "Similarity of neural network representations revisited." In International conference on machine learning, pp. 3519-3529. PMLR, 2019.

---

> ### Author Response · Authors · 2023-11-23
> **Response to Reviewer Dx4f - Q3, Q4, Q5, Q6, Q7**
>
> **Q3. Have you run experiments when training only the classifier?**
>
> [Smith et al., 2023] has conducted this experiment in their paper, where it is called "FT++".
> We quote those numbers in the below table and compare to OVOR.
> When only the classifier is trained, the resulting accuracy is significantly worse than OVOR.
>
>
> **Table**: Performance between FT++ and OVOR on different task numbers of ImageNet-R and CIFAR100.
>
> | | 5-task ImageNet-R | 10-task ImageNet-R | 20-task ImageNet-R | 10-task CIFAR100 |
> |---|---|---|---|---|
> | FT++ | 60.42 | 48.93 | 35.98 | 49.91 |
> | OVOR | 76.79 | 75.61 | 73.13 | 86.68 |
>
> Reference:
>
> - James Seale Smith, Leonid Karlinsky, Vyshnavi Gutta, Paola Cascante-Bonilla, Donghyun Kim,
> Assaf Arbelle, Rameswar Panda, Rogerio Feris, and Zsolt Kira. Coda-prompt: Continual de-
> composed attention-based prompting for rehearsal-free continual learning. In Proceedings of the
> IEEE/CVF Conference on Computer Vision and Pattern Recognition (CVPR), pp. 11909–11919,
> June 2023.
>
> **Q4. Is there an explanation for why the VOR increases forgetting?**
>
> In general, a method with higher accuracy might increase forgetting since the method have more room to forget.
> Although VOR increases forgetting, its increases in accuracy are larger than forgetting.
> As shown in the below table, where we report the start and final accuracy for each task on 10-task ImageNet-R.
> OVOR on average has higher starting accuracy than OnePrompt, which is used to compute forgetting, resulting in higher forgetting.
> Despite of this, OVOR is generally having higher final accuracy for each task as well.
>
> **Table**: Starting and final accuracy of OnePrompt and OVOR for each task on 10-task ImageNet-R.
>
> | Task index |1|2|3|4|5|6|7|8|9|10|
> |---|---|---|---|---|---|---|---|---|---|---|
> |OnePrompt (start)|91.90\%|79.75\%|83.18\%|72.89\%|81.29\%|79.56\%|72.17\%|75.31\%|73.94\%|71.13\%|
> |OnePrompt (final)|82.14\%|69.00\%|77.94\%|69.10\%|75.00\%|75.75\%|71.27\%|73.90\%|72.57\%|71.13\%|
> |OVOR (start)|92.27\%|82.68\%|83.73\%|78.86\%|85.00\%|84.31\%|75.58\%|82.23\%|73.08\%|71.13\%|
> |OVOR (final)|79.93\%|68.72\%|78.12\%|74.05\%|79.68\%|81.30\%|74.15\%|81.60\%|72.57\%|71.13\%|
>
> **Q5. In Section 5, you mentioned that for OnePrompt, you insert the prompts into the first 5 layers. Did you do an ablation of this hyperparameter?**
>
> Thanks for suggesting this ablation study. To fairly compare with previous methods, we follow the design of DualPrompt to insert the prompts without task-specific prompts.
> Here, we experimented with different configurations of prompts insertion and the results on 10-task ImageNet-R are shown in the following table (Table B.16 in the revision).
>
> **Table**: Ablation study of different prompt lengths and locations for OnePrompt.
>
> | Length 5 (start, end) | Length 20 (start, end) | Accuracy |
> |:---:|:---:|:---:|
> |(1,2)|(3,5)|73.34\%|
> |(1,12)| - |73.92\%|
> | - |(1,12)|74.69\%|
> |(3,4)|(3,5)|73.46\%|
> |(5,6)|(3,5)|73.41\%|
> |(6,7)|(3,5)|73.89\%|
> |(1,2)|(2,4)|72.89\%|
> |(1,2)|(4,6)|73.55\%|
> |(1,2)|(5,7)|73.93\%|
>
> The default option, which is the first row, is outperformed by many others, but except for the second and third one (inserting same-length prompts to all layers), the gains are negligible.
> The first row is used as the default because it is same as DualPrompt, and it is easier to compare with DualPrompt when OnePrompt has the exactly same prompt length and position.
> In the revision, we further add the results of the second row as OnePrompt-Deep in Table 3.
>
> **Q6. How costly is the VOR approach?**
>
> The virtual outlier only needs to be generated once for each task (not each epoch).
> As the exact computations depend on the size of the dataset, we measure the cost of VOR by the required extra training time of one task. For 10-task ImageNet-R, with VOR, it only takes extra 1.9\% increase in training time.
>
> **Q7. Do you have the results for OnePrompt for Table 4?**
>
> Yes, we have update that in Table 3 of the revision (Table 4 in the original paper).

---

### Official Review · Reviewer_2hZK · 2023-11-04

**Soundness:** 2 fair
**Presentation:** 3 good
**Contribution:** 2 fair
**Rating:** 6
**Confidence:** 2

**Summary:**

This paper considers rehearsal-free class-incremental learning (CIL), and proposes the "OnePrompt with Virtual Outlier Regularization" technique (OVOR) that performs equally or better than existing prompt-based CIL methods in terms of accuracy and computation. The proposed method consists of two elements, 1) the "OnePrompt" - a single prompt based mechanism that gives comparable results to mechanisms with multiple-prompts, 2) an outlier regularization technique that reduces the inter-task fusion by tightening the decision boundaries of the classes among different tasks.

**Strengths:**

- The proposed method for rehearsal-free CIL outperforms (or performs equally) existing methods in terms of accuracy, computation and storage.
- The pipeline is clearly explained

**Weaknesses:**

- The functionality of the virtual outliers is not very clearly explained.
- The paper lacks intuition on how the single prompt method works equally or better than the multiple prompt methods.

**Questions:**

- Is there any mathematical explanation/proof as to why the outlier method works well in tightening the decision boundaries, based on the Gaussian mechanism employed?
- Is there any intuition as to how why the single prompt method is comparable to the multiple prompt cases in terms of the performance?

---

> ### Author Response · Authors · 2023-11-23
> **Response to Reviewer 2hZK**
>
> We thank the reviewer for the comments, and we are glad that the reviewer finds our explanation of the pipeline clear. We address your concerns as follows:
>
> **1. Is there any mathematical explanation/proof as to why the outlier method works well in tightening the decision boundaries, based on the Gaussian mechanism employed?**
>
> As illustrated in Figure 1, the virtual outliers are used to prevent the classifier from producing high scores for samples in far away regions from training data in the feature space. Our idea is inspired by [Kim et al. 2022] which shows that if the feature encoder is more robust to out-of-distribution (OOD) samples, it reduces the inter-task confusion of CIL in general. On the contrary, instead of having the robust feature encoder, we use the virtual outlier samples to regularize the classifier head instead. The Gaussian noise added on the outer samples during the virtual outlier synthesis can be viewed as to guide the decision boundary of classifier head to have a margin from the outer samples. A theoretical proof of how the virtual outliers could tighten the decision boundary is one of our future direction.
>
> Reference:
>
> - Gyuhak Kim, Changnan Xiao, Tatsuya Konishi, Zixuan Ke, Bing Liu. "A Theoretical Study on Solving Continual Learning" NeurIPS 2022
>
> **2. Is there any intuition as to how why the single prompt method is comparable to the multiple prompt cases in terms of the performance?**
>
> Thank you for pointing it out where readers may find confusing, let us elaborate more on our intuition why OnePrompt can outperform DualPrompt, or on the other hand, why DualPrompt fails to provide better results.
> First, let us recap the design of DualPrompt.
> The core design of DualPrompt is have a prompt set, where each prompt has an 1-to-1 correspondence to one task.
> DualPrompt also design the prompt keys that are used to select proper prompt and the keys are likely converged to the mean representation of one task based on its minimization function.
> During inference, DualPrompt selects a prompt by finding the closest prompt key to the query image and then inserts the prompts to the ViT.
> Then, selecting the correct prompt ideally should be important to classification result for DualPrompt, otherwise it would contradict with the intent of the design.
> Nonetheless, given the composition of one task is purely random, there is no semantic relationship between mean representation of one task to a query image; thus, the matching rate of prompt is pretty low in DualPrompt. In their paper, on the 10-task ImageNet-R dataset, it only achieved **55.8\%** prompt matching rate on all test samples after trained with all tasks. Given the low match rate in DualPrompt, it would be limited how much the task-specific prompt mechanism can help the classification accuracy.

---

### Author Response · Authors · 2023-11-23
**General Response to Reviewers**

We would like to thank the reviewers for their time and effort in reading and commenting on the manuscript. We appreciate that the reviewers found the paper "**clearly explained**" (Reviewer 2hZK, DjrG) and **OnePrompt method is interesting and it surprised me how well it worked** (Reviewer Dx4f), and that our work's "**significance and originality**" (Reviewer DjrG) are acknowledged, along with our contribution in "**reducing computational cost**" (Reviewer Dx4f).
We have integrated all of the reviewers' major and minor comments and outlined changes that we have addressed in the revised version; the changes to the original main text and appendices are highlighted **in red** therein.

We summarize below the main improvements:

 - We clarify the experimental settings and the intuition behind our work, while providing more detailed explanation of why our method, with less computational cost, can achieve better performance than some of the baselines.
 - We give explanation for increased forgetting, and also clarify our remarks regarding prototype-based methods.
 - We share numbers that show that OnePrompt experiences little representation drift (Section E).
- We conducted additional experiments, on two new datasets, CUB-200 and ImageNet-A (Table 3), as well as the comparison of prompt-based methods with and without VOR (Table 2).
We demonstrate that VOR can universally improve the performance of all prompt-based methods, regardless of dataset.
 - We also performed ablation on the location and length of prompt (Section B).

Please feel free to add any additional questions or comments.

---

### Meta-Review · Area_Chair_RCxw · 2023-12-11

**Metareview:**

In this paper, the authors propose a novel method for prompt-based rehearsal-free class-incremental learning. After the rebuttal period, all reviewers agree to accept this paper. After checking the paper, reviews, and authors' feedback, I recommend acceptance. The authors are advised to include the discussion and revise the paper according to the reviewers' comments for the final version.

**Justification For Why Not Higher Score:**

All the reviewers give borderline acceptance. No one strongly supports this paper. Besides, this paper doesn't provide the results on some large-scale datasets, e.g., ImageNet-1k. The experimental settings are a little bit different from some popular continual learning papers, e.g., [iCaRL](https://arxiv.org/abs/1611.07725) and [LUCIR](https://openaccess.thecvf.com/content_CVPR_2019/papers/Hou_Learning_a_Unified_Classifier_Incrementally_via_Rebalancing_CVPR_2019_paper.pdf). Therefore, I think the paper is not of wide enough interest and should not be a spotlight or oral paper.

**Justification For Why Not Lower Score:**

This paper received consistent positive ratings after the rebuttal period.

---

### Decision · Program_Chairs · 2024-01-16

Accept (poster)